# Employment Management Policies for College Graduates under COVID-19 in China: Diffusion Characteristics and Core Issues

**DOI:** 10.3390/healthcare10050955

**Published:** 2022-05-22

**Authors:** Min Wu, Xinxin Hao, Yihao Tian

**Affiliations:** Department of Public Service Management and Public Policy, School of Public Administration, Sichuan University, Chengdu 610065, China; wuminhelen@163.com (M.W.); scuhxx@163.com (X.H.)

**Keywords:** COVID-19, graduate employment, policy diffusion, text analysis

## Abstract

The outbreak of COVID-19 epidemic has been having a great impact on the job market, so that graduates from all over the world are facing a more complex employment environment. Unemployment of the educated labor force often results in a waste of human capital and leads to serious economic and social problems. In the face of the impact of COVID-19, the Chinese government quickly introduced a series of employment policies for college graduates to relieve their employment pressure and create opportunities of career development. How did these employment policies for college graduates spread rapidly under the unconventional state of the COVID-19 epidemic? What are the diffusion characteristics? What are the core issues and measures? What are the differences between governments at all levels? These problems with rich connotation and research value needed to be further clarified. Based on the 72 employment support policies collected from the Chinese government network, this paper conducted a text analysis of the policies and found that in the process dimension, the employment policies of college graduates accumulated and exploded from bottom to top in the short term, and the policies diffusion followed the gradual model of “east–middle–west”. In the content dimension, there were five core issues: financial subsidies, innovation and entrepreneurship to drive employment, public institutions to absorb, optimizing public services, and lowering the support threshold. Meanwhile, there were obvious differences in the choice of policy tools, policy intensity, and implementation ideas in each region. The findings are of important significance for developed and developing countries to better respond to the impact of various emergency situations.

## 1. Introduction

The sudden outbreak of coronavirus disease 2019 (COVID-19) in early 2020 has dramatically changed people’s lives. Almost all industries were shut down and production stopped. As a result, the demand for labor declined in all sectors of a country. Under the background of the COVID-19 epidemic, the imbalance between supply and demand of the employment market has a great impact on the fresh graduates who are about to step out of campus and enter the society [1]. Unemployment of the educated labor force often results in a waste of human capital and leads to serious economic and social problems, and thus has attracted much of the attention of policy makers and researchers [2]. In a crisis situation, active labor market policies can increase the probability of signing a job contract in the job market [3]. 

According to the China Bureau of Statistics, the number of college graduates in 2020 in China reached 8.74 million, an increase of 400 thousand over the previous year, the largest number of graduates in history. While doing its job in preventing and controlling the epidemic, the Chinese government introduced a series of employment policies for college graduates in a short period of time in order to promote the stable employment of graduates, and the unitary state power structure enabled employment policies to spread rapidly from the local to the national level [4]. The data of the Chinese Ministry of Human Resources and Social Security show that, with the outbreak of COVID-19, the unemployment rate in China also stabilized at 4.4% in 2020. For these reasons, the findings of this study reference how governments around the world can help college students survive the cold winter of employment during the COVID-19 epidemic and other emergency situation.

At present, studies on employment policy in emergency situations have mainly focused on how to implement active labor market policies during the financial crisis [5,6,7], and there is little research on how to implement active employment policy under the emergency of a COVID-19 epidemic situation. Regarding how to deal with employment dilemmas during the epidemic, most studies focus on public health human resources and other specific groups [8,9], with less attention paid to the group of college graduates. Most studies on policy diffusion follow the gradual diffusion model, that is, with the passage of time, the cumulative adoption frequency of policy innovation follows the general process of “slow growth at the beginning-rapid increase in the middle-slow saturation at the end” [10,11]. Few studies pay attention to the characteristics and laws of policy diffusion in a short period of time.

In general, the existing research focuses little on the employment policy of college graduates under the emergency state of the COVID-19 epidemic situation, especially the lack of real exploration of the diffusion characteristics and core issues of the policy under the emergency state. Compared with the existing literature, the marginal contribution of this paper is to explore the diffusion characteristics of the government’s policy to promote the employment of college graduates under the emergency state, to analyze the core issues and main measures of the rapid diffusion of employment policies for college graduates under unconventional situations, and to analyze the preference for the use of policy tools to promote the employment of college graduates in various regions on the basis of provincial documents. 

In the sections below, we first present a review of scholarship on policy diffusion and employment policy. While we acknowledge these studies’ contributions, we pinpoint a gap that our research seeks to fill: the diffusion of employment policies for college graduates in emergency situations. Second, we explicate our research and design, including case selection, data collection, and research approach. Finally, we present our findings, limitations, and reflections on our conclusions.

## 2. Literature Review

### 2.1. Employment Policy

Employment policy is the policy measure and institutional arrangement adopted by a government to solve the employment problem in different periods. It not only includes improving the employment environment, expanding employment, and optimizing the economic structure, but also includes solving specific unemployment and other problems, such as job introduction, vocational training, and so on. In the existing public employment policy research, the scholars discuss the evolution path of specific employment policy, the development course of specific employment policy, the evaluation of specific employment policy, and so on. In view of our research theme, this article mainly focuses on why a government needs to actively implement employment policy to intervene in the labor market, and the specific means of intervention.

The reason why a government needs to actively implement the employment policy is that, on the one hand, the positive changes in employment are the result of the interaction between favorable economic trends and large-scale employment promotion measures [12]. Carr [13] believes that a generous policy support will cushion the negative impact of job insecurity by reducing the difficulty of finding similar jobs or providing income maintenance during unemployment. Jochen et al. [14] identified all relevant empirical studies with rigorous evaluations produced over the last ten years. The results of the analysis showed that youth employment programs such as skills training, entrepreneurship promotion, employment services, and subsidized employment had a statistically significant positive effect on employment promotion. On the other hand, a study showed that a history of poor employment between the ages of 25 and 50 was associated with poor health in the later stages of life, in particular recurrent unemployment, involuntary unemployment, weak labor market linkages and disadvantaged occupations [15]. Previous research also quantified the cumulative negative effects of multiple barriers to employment and proved the correlation between individual stress factors and labor market indicators. For every standard deviation in the number of barriers to employment, the probability of unemployment increased by 2.6 to 3.1 percentage points [16].

Based on the research results of most scholars, the strategies that can be considered in the process of implementing employment policy can be summarized as follows: firstly, promote the integration of interregional employment policy [17] from a unified employment strategy, ensure that the active employment policy becomes a national policy priority [5], and ensure the continuity of the employment strategy and the regulation of the implementation of employment strategies [18], thereby promoting the development of the field of employment. Secondly, we should improve employment laws and policies, strictly implement the minimum-wage system, and increase the opportunity cost for enterprises to fire employees [19], guarantee the employment rights and interests of disadvantaged groups in terms of wages, training and promotion [20], implement generous labor market policies, use fiscal incentives such as taxes, subsidies and transfer payments to promote career choice, and create opportunities for integration into society and labor for young people between the ages of 18 and 24 [21]. Employment discrimination on the basis of education, major, sex or disability is widely prohibited, and good labor market conditions are created [22]. Thirdly, innovate in social services, promote enterprises to improving the working environment, create a good workplace environment for the employed [23], reduce the costs of transportation, housing, medical care and other aspects of employment [24], improve public finance, and promote the joint provision of relevant employment services by the public and private sectors [25]. Fourth, based on the fact that employment is divided into guaranteed employment and nonstable employment [26], some scholars believe that the government should pursue a labor market policy to promote the full employment of internal and external personnel, so as to increase the employment rate and reduce welfare dependence [27]. Most studies have found that self-employment policies have a positive impact on employment status and personal income [28], so flexible labor markets should be developed to promote informal employment [29] and flexible employment [23], such as temporary employment and part-time employment [30], and take positive measures to encourage self-employment and create jobs [31]. Finally, there is a need for scholars to take measures to encourage higher education institutions to pay attention to improving the key employability of graduates in order to enhance the employment resilience of graduates after entering the labor market [32].

### 2.2. Employment Policy in Emergency Situations

There is little research on employment policy in emergency situations, mainly focusing on how to implement active labor market policies during a financial crisis [5,6,7], and very few about major natural disasters, such as employment assistance that should be implemented after earthquakes [33]. At present, there are many research results on how to deal with the employment dilemma during an epidemic, most of which are about how to promote public health human resources [8], medical students [34,35,36], nurses [37,38], air service personnel [9], and other specific groups. In addition, Tinggui et al. [39] analyzed the social effects of employment promotion policies for college graduates under COVID-19 by using the PMC index model. The existing literature pays less attention to the employment of college graduates, and the research on the employment policy of college graduates under the emergency situation of COVID-19 is even scarcer, which also provides an opportunity for us to study the employment policy and its diffusion for college students in the country under the emergency situation.

### 2.3. Policy Diffusion

Policy diffusion is the abbreviation of policy innovation diffusion, which usually refers to the process in which a policy scheme is transmitted from one department or region to another and adopted and implemented by the new policy subject [40]. Policy diffusion research began in the 1960s. Policy diffusion has been a focus topic in the field of social science for decades [41]. After more than 50 years, policy diffusion theory has become an important theory of policy process research, which has been verified and applied in many countries [42,43,44]. 

Policy diffusion research can be divided into two categories: “variable research” and “process research” [11,45]. “Variable research”focuses on “why policy innovation spreads (Why)”, that is, it mainly focuses on the influencing factors of different policy adoption behaviors and results, and most of them are quantitative studies [46,47]. “Process research” pays attention to “how to spread policy innovation (How)”, that is, it analyzes the time series of events to examine the core dynamics and mechanisms of adoption decisions and policy diffusion and is both quantitative and qualitative [48].

With regard to how policy innovations diffuse (How), influenced by decision models such as limited rational decision-making and progressive decision-making, researchers have argued from the outset that “the policy adoption process is logically consistent with incrementalism” [49]. Research on the policy-making process shows that state officials often look for clues and inspiration for state policymaking from decision-making programs that have proved effective or promising [50]. In this process, adoption decisions are made through information-seeking and information-processing behaviors, which determines that policy diffusion is a gradual decision-making process when decision makers face complex constraints such as time constraints, cognitive ability constraints, and uncertainty [51]. Empirical studies of policy diffusion in a range of areas such as agricultural development, education, welfare, medicine, and democratic rights have also repeatedly shown that cumulative policy adoption frequency follows a “slow growth at the beginning-rapid increase in the middle-slow saturation at the end” over time [52]. The S-shaped diffusion curve is therefore considered a general feature of the policy diffusion process. 

However, the diffusion curve of policy innovation between states does not always follow the pure trajectory (S-shaped curve) respected by traditional policy diffusion research. We need both a better theory and better tests to determine whether the S-shaped curve, so frequently discussed in the literature, represents an informative substantive phenomenon [53]. Boushey summarized a variety of nonprogressive policy diffusion curve forms, such as “steep S-type, R-type and ladder”. The differences in the distribution and peak state of the adopted data presented by these atypical policy diffusion curves prove the possibility of the nonprogressive policy diffusion model. Based on the research results of an epidemic transmission mechanism, Boushey used the concept of “Policy Outbreaks” to describe the extremely rapid policy diffusion process in which the diffusion curve presents exponential characteristics and constructed the epidemiological analysis framework of the dynamic mechanism of policy diffusion in the United States [54]. A few scholars have also demonstrated a nonprogressive policy diffusion pattern, with Easterly [55] noting that the diffusion process of the Sex Offender Registration and Notification Laws (SORNL) deviated from its normal trajectory by experiencing rapid diffusion in a short period of time, showing an intermittent diffusion. Douglas et al. [56] showed that the involvement of top-down go-betweens greatly increased the likelihood of coercion as a mechanism for policy diffusion, thus giving rise to a policy explosion of “drug courts” after a period of slow diffusion. In general, scholars still have relatively little research on explosive policy diffusion.

Therefore, compared with the policy diffusion in conventional situations, how did the employment policy for college graduates spread rapidly in the unconventional state of the COVID-19 epidemic? What are the diffusion characteristics? Do the core issues and measures accurately connect with the needs of college graduates? What are the differences between governments at all levels? What are the areas that need to be further improved and strengthened? These problems with rich connotation and research value need to be further clarified, in order to better deal with the impact of emergencies all over the world and do a good job in the employment of college graduates to provide experience and reference.

## 3. Research Design

### 3.1. Data Source

The study used data from the “stable Employment support Policy” special index database on the Chinese government website. The database is a special index database of the “stable Employment support Policy” launched by the Chinese government on the Chinese government website in January 2020 to help unemployed people easily access and make full use of employment support policies, which includes the relevant support policies issued by the central and local governments to promote employment. On the basis of this database, “graduates” were selected in the identity retrieval column, and “employment assistance” or “internship employment” were selected in the query content column, and a total of 186 documents issued by the central and provincial levels were retrieved. In order to accurately determine the research object, the 186 documents mentioned above were screened in two steps. The first step was to screen out the policy of “promoting the employment of college graduates” in the title of the document or the title of the first level, to ensure that the policy content was in line with the purpose of the research; the second step, on the basis of the first step, was to screen out the policies issued after January 2020, so as to accurately locate the employment support policy in the state of emergency. A total of 72 valid documents were obtained finally after screening. At the same time, all employment support policies that had been introduced were checked by hand to ensure accuracy. In order to ensure the comparability among the subjects and avoid cross-level comparison, 72 documents were classified, and 4 valid documents at the central level and 68 valid documents at the provincial level were obtained, which were distributed in 28 provinces, municipalities, and autonomous regions. This paper took 28 of the 72 documents as the basis of the quantitative analysis of the policy literature, and the special or measures documents as the supplement.

### 3.2. Research Approach

Policy literature measurement is a quantitative research method to analyze the structure of the policy literature, which originates from bibliometrics; it introduces sociology, mathematics, statistics, and other subject methods into policy analysis and reveals the cooperation mode of policy subjects, as well as the structure and evolution of the policy system in a visual way.

In this paper, a time series analysis and cluster analysis were used as the main analysis methods, and NVIVO 11 plus (QSR International, Melbourne, Australia) was used as the basic analysis tool. Firstly, the publishing subject and time of the 72 documents were counted, and the policy release time of different subjects, different levels, and different subjects at the same level was compared, and the trend and hierarchical characteristics of policy diffusion were explored from the perspective of time and space. Secondly, the outline policy was introduced into NVIVO 11 analysis software, and the five core issues of the policy were obtained through coding and a cluster analysis. Finally, we used the software’s internal correlation and matrix query results to analyze the heterogeneity of different regional policy tools.

## 4. Analysis of Policy Diffusion Process

The analysis and discussion of the specific process of policy diffusion is the key content in the study of policy diffusion, which has many research dimensions, such as how the policy spreads in time, how to distribute it in space, how to transmit it at a level, and so on. This paper discusses the diffusion characteristics of the employment policy of college graduates in the emergency state from a time dimension, a hierarchical dimension, and a spatial dimension.

### 4.1. Time Dimension: Policy Accumulates and Explodes in the Short Term

Generally speaking, the introduction of a policy requires a long period of preliminary research, source innovation, and learning adoption. Different from the traditional policy diffusion model, the policy of promoting the employment of college graduates in the emergency situation erupted rapidly in a short period of time, showing the characteristics of accumulation and diffusion, as shown in Figure 1. From January to February 2020, there was a surge of 23 provincial policy documents, with 61 percent of provincial governments issuing documents in February and 83 percent of policy documents issued within 20 days in February, meaning that most provincial governments adopted the policy within a month. It can be judged that the spread of policy documents to promote the employment of college graduates during the epidemic situation is a typical “policy outbreak”. The essence of the accumulation and outbreak of the policy in a short period of time is the imbalance of the government attention under the condition of limited rationality under conventional conditions; when there are sudden factors affecting social stability, the government quickly shifts from conventional management to governance related to an emergency situation, such as an epidemic situation, which affects people’s livelihood.

### 4.2. Hierarchical Dimension: Nonvertical Flat Diffusion Absorption

In order to analyze the hierarchical trend of policy diffusion, this paper took the time of the first policy support document issued by the local and central governments to promote the employment of college graduates as the key node during the epidemic situation. The provincial level policy was issued by the Jilin Provincial Department of Human Resources and Social Security on 22 January, that is “The Opinions of Jilin Provincial Department of Human Resources and Social Security, Department of Education, Public Security Department, Department of Finance and the Changchun Central Branch of the People’s Bank of China on the Implementation of the Employment and Entrepreneurship Work of College Graduates under the Current Situation”; and the national level policy was issued by the Ministry of Education of the People’s Republic of China on 4 March, that is “The Notice of the Ministry of Education on Responding to the Epidemic Situation of COVID-19 to Do a Good Job in the Employment and Entrepreneurship of the 2020 National College Graduates”; According to the comparison of release time, the release of relevant documents to promote the employment of college graduates began with the local government, and then gradually spread to other provincial governments. In total, 67% of the provincial governments issued policies before the central government adopted the relevant policies on 4 March. Under the emergency situation, the formulation and diffusion of employment policy for college graduates did not come from the external pressure brought about by the high promotion of the central government, but from the internal demand of provincial (municipal) local governments.

In the path of policy diffusion, different from the vertical top-down implementation of the policy, the employment support policy for college graduates began to be implemented by the local government, and the local government gradually drew lessons from it and adopted it. Before it rose to the will of the central government, the policy had spread to most provincial governments, reflecting the characteristics of a nonvertical flat diffusion of the employment support policy for college graduates. The characteristics of these hierarchical dimensions show that the intrinsic driving force of policy formulation in the state of emergency was at the local level. The local colleges and universities were directly facing the provincial governments, and the predicament and demand of the graduates’ employment urged the provincial governments to issue the support policy. Secondly, before the peak employment of college graduates, the local governments predicted the impact of the COVID-19 epidemic situation and responded quickly, effectively saving the diffusion time needed for the spread of a top-down policy, helping to implement the policy quickly, and reducing the distractions of the central government at the same time.

### 4.3. Spatial Dimension: Progressive Spatial Diffusion of “East–Middle–West”

From a spatial dimension, the diffusion of public policy is a gradient, which shows that a policy diffuses from regions with higher policy potential to regions with lower policy potential, which gives birth to the proximity effect and agglomeration effect and promotes the policy to show the gradual diffusion trend of a cluster. From the employment policy point of view, although most provincial governments issued the relevant policies in February, it seems to exhibit the characteristics of multipoint diffusion. However, while the region was sorted according to the time that each province issued the first policy to promote the employment of graduates (Figure 2, the horizontal axis indicates the time of release, the vertical axis indicates the number of releases, and the publishing area is marked on the column chart for short), from 22 January to 21 February, after excluding Jilin (Ji) and Qinghai (Qing) provinces, the diffusion sequence of this policy was consistent with the diffusion order of the economic policy in space, which was manifested in the diffusion from east to west, accompanied by a cluster effect and a proximity effect.

The reason is that the economy restricts the development of higher education, and the development of colleges and universities depends on the local economy. Therefore, the quantitative distribution of colleges and universities is closely related to the local economy. The concrete performance of the number of colleges and universities is eastern > central > western. There was an urgent need to formulate support policies to solve the employment problem of college graduates in the eastern provinces, which have a large number of colleges and universities. The urgency of policy formulation in areas with a small number of colleges and universities was slightly less than that in economically developed areas. Therefore, the diffusion order of the employment policy for college graduates was consistent with that of the economic policy.

## 5. Core Policy Issues

The policy text analysis based on NVIVO relies on the rooted theory of the postpositivism paradigm, looks for the core concepts that reflect the essence of a phenomenon on the basis of the systematic collection of data, and then constructs the relevant social theories through the relationship between these concepts, which can reflect the core issues, policy subjects, and objectives of a policy from many aspects. In policy research, a cluster analysis is an important tool to excavate the structure and significance of policy text. In this paper, the programmatic policies of 28 provinces were introduced into NVIVO 11 analysis software. A total of 186 nodes, 129 subnodes and 214 reference points were obtained by coding, and a total of 21 nodes were obtained after collation. A cluster analysis was carried out according to the meaning of the nodes, and a tree diagram of the cluster analysis results is shown in Figure 3. The closer the distance between the nodes, the higher the correlation between the two nodes. For example, the distance between “retaining the identity of the graduates” and “Internet + service” is close, indicating that the Pearson correlation coefficient between the two nodes is large and the correlation degree is high, which belongs to the same core topic. Combined with the results of the cluster analysis, this paper summarized and divided the core issues of the target policy into five categories.

### 5.1. Increase Financial Subsidies

In accordance with the law of the People’s Republic of China on the promotion of employment, the governments at or above the county level shall, in accordance with the employment situation and the objectives of the employment work, allocate special funds for employment in the financial budget for the promotion of employment. As an important control tool to promote employment and stabilize people’s livelihood, how to give full play to its positive role in promoting employment in an emergency situation is the most important work of the provincial governments. It was found that in addition to Guangxi, Henan, Inner Mongolia, Xinjiang Construction Corps, the other 24 provinces issued relevant subsidy policies, and a total of 74 subsidy terms or rules appeared in the text library. Through a semantic analysis of the policy text, the relevant nodes of the subsidy policies were summarized and sorted out. As shown in Table 1, the four subnodes of the subsidy policies were: subsidies to graduates, subsidies to enterprises, subsidies to trainee units, and subsidies to third parties (employment service institutions). It means that the policymaking subject pays subsidies to the four subjects involved around the supply and demand sides from an actual point of view. Compared with the conventional employment promotion policy, the scope of subsidies in the emergency state was broader, and the emphasis of subsidies was tilted from graduates to employers and third parties.

The primary recipient of subsidies was college graduates, and there were 34 subsidies for college graduates in the policy text. Subsidies to graduates enhanced the protection of their basic life, gave them spiritual and material support, and ensured that college graduates can focus on finding employment opportunities. The types of subsidies included personal skills upgrading subsidies, job search and entrepreneurship subsidies, and so on. At the same time, some provinces, such as Gansu Province, carried out special project subsidies under the existing central subsidy policy for the employment in emerging industries of college graduates with a living allowance of 1500 yuan per person per month for a period of three years, to promote the development of emerging industries and employment.

The second recipient was the enterprise. Under the influence of the COVID-19 epidemic situation, most enterprises stopped work and stopped production, and the demand for employment was reduced, which led to the reduction of the employment market of recent graduates. As the main factor affecting the employment of college graduates, the provincial governments, on the basis of issuing a large number of policies to support small- and medium-sized enterprises, at the same time, provided employment subsidies for enterprises as the main means to promote college graduates. According to the coding results, 14 provinces, such as Shanghai and Guangdong, subsidized enterprises, and 21 provisions on subsidies appeared in the target policy documents. It can be said that some provinces emphasized subsidies with enterprises many times.

Finally, it subsidized the third party (employment service) institutions to promote employment and the trainee units that provide employment opportunities, make use of the third-party platform resources to improve the employment ability of college graduates, dock the supply and demand sides to transform the importance of employment from a document level to a substantive operation, and implement the employment assistance to college graduates.

The subsidy policies not only showed that the provincial government’s attention to resolving the employment pressure of college graduates was transformed into substantive support to improve the popularity, guarantee, and incentive function of the policy, but also further showed that the adaptability of the local subsidy system to unconventional emergencies was constantly improving, and the mechanism of alleviating employment pressure and stimulating employment vitality was constantly maturing.

### 5.2. Promoting Employment through Innovation and Realizing Employment through Entrepreneurship

In the unconventional emergency situation, in the face of the largest scale of college graduates in history, innovation and entrepreneurship has become a new engine to promote employment. The analysis found that 11 provinces, including Beijing, Guangxi and Fujian, encouraged graduates to innovate and start their own businesses, and a total of 22 detailed rules or provisions supporting innovation and entrepreneurship to promote employment appeared in the policy text.

Table 2 shows the specific measures to promote employment by innovation and entrepreneurship mainly included: incubation service, innovation and entrepreneurship training, innovation and entrepreneurship education, entrepreneurial loan guarantee. For example, Beijing not only used Zhongguancun Entrepreneurship Street and Municipal Entrepreneurship Park to provide professional services for college students, but also supported colleges and universities to hold innovation and entrepreneurship competitions and select potential projects to hatch. Innovation and entrepreneurship training and education are important ways to promote innovation and start a business so as to promote employment and help graduates to achieve employment more actively and fully. The four governments of Beijing, Tianjin, Guangxi and Chongqing also reduced the conditions for applying for entrepreneurial loans for graduates. For example, in addition to supporting college graduates to apply for loans as a priority, Chongqing also established a credit village, a credit park, and a career fair for entrepreneurship incubation to recommend a guarantee-free mechanism. The incubation service and entrepreneurial loan guarantee policy promoted the transformation of high-quality projects and greatly improved the possibility of entrepreneurial success, indicating that in the unconventional emergency state, the mechanism to resolve employment risks and pressures could respond quickly to demand. However, local governments were insufficient in combining their own industrial advantages and regional development characteristics to drive development innovation, which reflects that the attention distribution of local governments needs to be balanced, and emergency policies need to be refined and individualized. Different from the conventional policy to promote the employment of graduates, a series of measures to reduce the cost and risk of entrepreneurship under the emergency state, such as the cancellation of antiguarantee and the recommendation of guarantee-free mechanism, were more effective to stimulate graduates’ awareness of innovation and entrepreneurship. 

### 5.3. Promoting State-Owned Enterprises and Public Institutions Absorb College Graduates

The government attaches great importance to give full play to the exemplary leading role of public institutions and state-owned enterprises and promotes the employment of graduates through flexible and ingenious job creation. Through the coding analysis of the target policy, it was concluded that the provincial government mainly promoted graduates to the local employment needed by the state through four main ways shown in the Table 3: encouraging college students to enlist in the armed forces, employment at the grass-roots level, adding scientific research assistant posts, and absorbing students in state-owned enterprises. Compared with the conventional policies to promote the employment of graduates, it was found that under the emergency conditions, posts of scientific research assistants were added, a number of policy documents were issued to give full play to the strength of state-owned enterprises to promote the employment of graduates, and special job fairs for state-owned enterprises were held.

The employment policy at the grass-roots level combines the employment demand of college graduates with the goal of building a well-off society in an all-round way, encourages more college graduates to serve the people at the grass-roots level and build the motherland, which effectively slows down the employment pressure of college graduates. The enlistment of college students is one of the important means to promote the high-quality development of the army and realize the national defense power. In order to solve the employment risk, some local governments also formulated specific growth targets for the enlistment of college students, such as the Jiangxi government, which issued a policy “encouraging more college graduates to enlist in the armed forces, and strive to achieve the goal of recruiting more college graduates from the same period last year”. At the same time, state-owned enterprises were the main target for graduates to achieve employment. Fifteen local governments, such as Shanghai and Jiangsu, introduced policies to encourage state-owned enterprises to create flexible posts, and to subsidize employment enterprises that had expired their internships. Scientific research assistant refers to the research assistant post established with project funds as the main source of funding. Beijing and Tianjin took the lead in issuing policies before 1 July to support the establishment of scientific research assistant posts, alleviate the pressure of employment, and promote the scientific research and development of colleges and universities and scientific research institutes.

### 5.4. Optimizing Public Services

Since the college graduates choose their jobs, the Chinese government provided professional and convenient public services for college graduates through the combination of supply and demand, the integration of various channels and the rational allocation of resources, such as signing third-party agreements and amending the Labor Protection Law of the People’s Republic of China, which not only satisfy the employment of enterprises, but also protect the employment rights of graduates. In an emergency situation, the public service should respond quickly, make flexible changes, innovate in the way the service is provided and in the content of the service, and effectively serve the needs of each graduate. As shown in Table 4, simplifying the employment process, retaining the status of recent students, and “Internet + services” highlight the discussion on the theme of public service policy. Regarding the way the service was provided, Heilongjiang was the first to put forward the use of the Internet to carry out online recruitment, followed by Beijing, Hubei, Hebei, and other areas to refine the policy details of “Internet + services”.

As far as the content of the service is concerned, the local governments mainly reduced the employment time cost of employers and college graduates through the simplification of government and decentralization. Compared with the traditional public service policy, it was more convenient for employers and graduates to vigorously promote online double election and online employment and other services. At the same time, some local governments, such as Beijing, issued a supportive policy to retain the status of unemployed graduates for two years, which effectively alleviated the risk that the employment lag of college graduates brings to the society.

### 5.5. Lower the Threshold of Employment and Help with Precision

For the difficult students whose competitive advantage is not significant in employment, the provincial government established a support mechanism for graduates who had not obtained the vocational qualification certificate on time under the influence of the COVID-19 epidemic, relaxed the entry threshold in stages, played a significant role in promoting employment, and ensured that college graduates could achieve better and fuller employment. For those with employment difficulties, such as poor households’ graduates, disabled graduates, the establishment of accounts, through vocational training, job recommendation, and other means, was provided to achieve one-to-one employment guidance and support. In addition, combined with the goal of a high-quality development of higher education, to promote the development of higher education to the people, Heilongjiang, Hunan, Fujian, and other nine local governments issued fresh graduate education policies; for example, Hunan Province issued a policy to “increase the proportion of college students more than doubled in 2020”. In addition, Shanghai, Tianjin, Fujian, Chongqing, and other places also formulated specific enrollment expansion places to effectively implement the policy of further study for fresh graduates and ease the pressure on employment.

Some professional and technical practitioners, such as primary and secondary schools, kindergartens, teachers, patent agents, and other professions, need to obtain the corresponding qualification certificates in order to be able to engage in their profession. Affected by public health emergencies, the examination of vocational qualification certificates could not be held on schedule. Some local governments adopted the policy of relaxing the threshold of vocational access in stages, such as the vocational qualifications of primary and secondary schools, kindergartens and secondary vocational schools issued by the local government in Guangxi. The teachers were allowed to get the qualification certificate after inauguration, so that it could reduce the difficulties brought about by the emergency.

## 6. Discussion

This study explored the diffusion characteristics and core issues of employment policies in emergency situations by means of a time series and cluster analysis and so on. Further, the study analyzed the heterogeneity of policy in various localities from the attention allocation difference to the choice of policy tools and explored the policy implications.

### 6.1. Attention Allocation Difference

In the face of emergencies, the reasonable allocation of government attention is the premise of effectively resolving the employment pressure of college graduates. The difference in attention allocation between local governments was reflected in the enthusiasm and attention of the policy. From a positive point of view, the Jilin Department of Human Resources and Social Security, under the leadership of the government, took the lead in issuing measures to promote the employment of college graduates under the leadership of the Provincial Education Department, the Public Security Department, the Department of Finance, and the People’s Bank of China. On 21 April, the Jiangxi People’s Government issued a letter as the last of the 28 provinces. The 90-day interval between Jilin Province and Jiangxi Province shows that there are differences in the response enthusiasm of each province to the employment pressure of college graduates under the condition of emergency. In terms of attention, as of 1 July 2020, six administrative districts of Inner Mongolia Autonomous region, Hunan, Guangxi, Guizhou, Guangdong, Ningxia, and Xinjiang Construction Corps have issued only one outline document, while Shaanxi, Shandong, Fujian, Tianjin and other administrative regions have issued more than four relevant policies, which span from February to June, indicating that during this period some governments continued to pay attention to the employment of college graduates, and establish relevant supporting programs to ensure that the policy could have a stable impact. However, some governments have only issued programmatic documents.

### 6.2. Differences in Preferences for the Choice of Policy Tools

There were also differences in the choice of policy tools among provinces. As shown in Table 5, the policy preferences in the eastern region are subsidies, institutions absorption, and accurate help in turn; the northeast region prefers to use subsidies, precision help, and optimize employment services; the middle region focuses on the absorption of institutions, followed by accurate help, and finally subsidies; the western region focuses on the use of subsidies, followed by institutions, and finally optimizes employment public services. It is not difficult to see that a subsidy policy, as a preferential policy, can directly alleviate the employment pressure of college graduates in a short period of time, so it can be used as the first-choice tool in the eastern, northeast, and western regions. In addition to subsidies, there are significant differences in the preferences of the provinces for the use of policy tools. The eastern and western regions show a strong help for the absorption of public institutions, and the northeast and the middle regions advocate the promotion of accurate help. The western region has the least adoption of employment promotion policies, while the eastern, northeast, and middle regions have the least adoption of phased measures.

### 6.3. Policy Implications

This research adds to the existing literature by exploring the diffusion of employment policies for college graduates in emergency situations. The findings suggest that an intrinsic driving force of policy formulation in the state of emergency is at the local level of government. This may be because even in some countries with unitary systems, the subnational governments could enjoy certain economic or administrative autonomy. The findings of this study have implications for how governments around the world can help college students navigate employment difficulties to have better career development opportunities during the COVID-19 epidemic and other emergency situation. Meanwhile, the findings provide evidence on how countries should rapidly diffuse policies to respond to negative situations in emergencies and have important reference implications for how countries can mitigate the negative impacts of various emergencies.

## 7. Conclusions

Generally speaking, in an emergency situation, local governments were able to quickly respond to the employment pressure of college graduates, form countermeasures from the bottom up, and actively mobilize the strength of all parties and accurate policies to resolve the employment pressure of college graduates from both supply and demand. From the policy characteristics, the accumulation of explosive growth in a short period of time, the level of nonvertical flat diffusion absorption, the spatial diffusion characteristics and economic policy diffusion characteristics were consistent and formed the core issues in five aspects. At the same time, in the choice of policy tools, the preference of using policy tools had obvious regional characteristics. However, from the interpretation of the policy in this paper, we can see that the policies issued by local governments lacked innovative measures, did not combine employment with local emerging industries, and how to implement local policies lacked supervision. This research still had the following limitations, which need further study. First, the research object of this paper was the employment support policies introduced by the government for college graduates from January to July 2020. The impact of the COVID-19 epidemic is not yet over, and the government will issue new employment promotion policies. What are the future trends of employment promotion policies? How effective are the policies? A further analysis of the effects of the new policies is required. Second, this paper analyzed the diffusion of employment policies during emergencies based on a policy database in China. However, COVID-19 has impacted all countries in the world. A comparative study using data from different countries will be an important research direction in the future.

## Figures and Tables

**Figure 1 healthcare-10-00955-f001:**
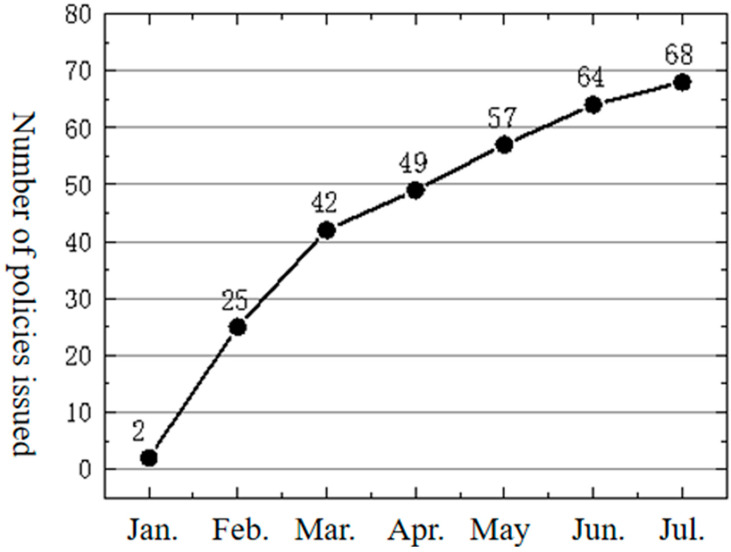
Trends in policy issuance.

**Figure 2 healthcare-10-00955-f002:**
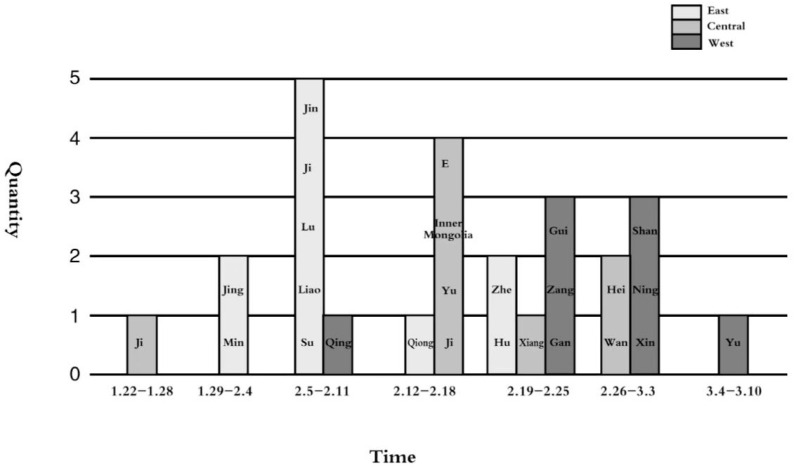
Spatiotemporal map of policy release. Ji represents Jilin Province; Jing represents Beijing; Min represents Fujian Province; Jin represents Tianjin; Ji represents Hebei Province; Lu represents Shandong Province; Liao represents Liaoning Province; Su represents Jiangsu Province; Qing represents Qinghai Province; Qiong represents Hainan Province; E represents Hubei Province; Inner Mongolia represents Inner Mongolia Autonomous Region; Yu represents Henan Province; Jin represents Shanxi Province; Zhe represents Zhejiang Province; Hu represents Shanghai; Xiang represents Hunan Province; Gui represent Guizhou Province; Zang represents the Tibet Autonomous Region; Gan represents Gansu Province; Hei represents Heilongjiang Province; Wan represents Anhui Province; Shan represents Shaanxi Province; Ning represents Ningxia Autonomous Region; Xin represents the Xinjiang Uyghur Autonomous Region; Yu represents Chongqing.

**Figure 3 healthcare-10-00955-f003:**
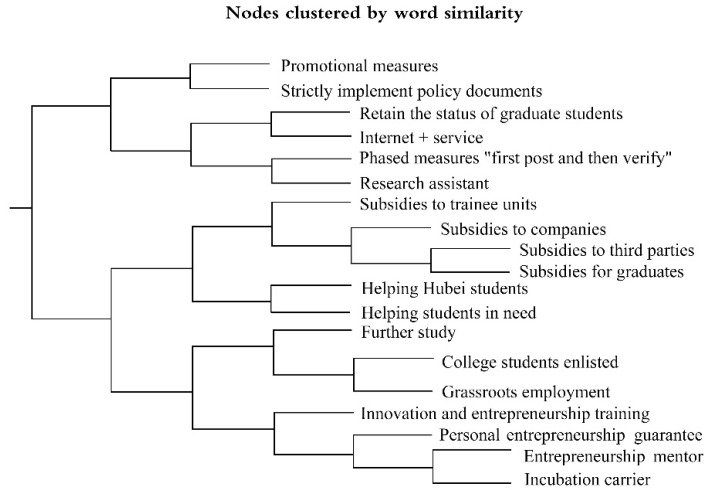
Cluster analysis result diagram.

**Table 1 healthcare-10-00955-t001:** Subsidies.

Type	Node	Reference Point	Typical Description
Subsidies to graduates	13	34	College graduates employed by enterprises may apply for personal-skills-upgrading subsidies in accordance with regulations.
Subsidies to enterprises	14	21	If small- and medium-sized enterprises absorb college graduates and pay social insurance in accordance with the regulations, they may be given employment subsidies according to the standard of 1000 yuan per person.
Subsidies to trainee units	10	10	For those who retain trainee personnel in trainee units, a retention reward of 3000 yuan shall be given to each person retained.
Subsidies to third parties	3	7	For those who set up an enterprise within half a year after training, they shall, in accordance with the standard of 1000 yuan per person, be given a subsidy for the success of starting a business in a training institution.

**Table 2 healthcare-10-00955-t002:** Innovation and entrepreneurship drive employment.

Type	Node	Reference Point	Typical Description
Incubation service	5	7	Relying on Zhongguancun Entrepreneurship Street and Municipal College Students Entrepreneurship Park, “one Street, three Parks, more points” incubation career.
Innovation and entrepreneurship training	4	5	Special training programs for key groups such as college graduates, retired soldiers, the poor and disabled, etc.
Innovation and entrepreneurship education	5	6	Promoting entrepreneurship, promoting employment, strengthening innovation and entrepreneurship education, integrating professional education and innovative entrepreneurship education in depth.
Entrepreneurial loan guarantee	4	5	Priority should be given to supporting college graduates in entrepreneurship guarantee loans, entrepreneurial rent subsidies, entrepreneurial incubation, and so on.

**Table 3 healthcare-10-00955-t003:** Absorption measures of public institutions.

Government-Affiliated Institutions	Node	Reference Point	Typical Description
College students enlist in the army	16	18	Increase the proportion of college students enlisting and give priority to approving the recruitment of college graduates into the armed forces.
Grass-roots employment	22	20	Expand the scale of employment at the grass-roots level, continue to do a good job of rural teachers’ special post plan, rural revitalization of the special recruitment of associate staff.
Absorption by state-owned enterprises	15	13	Provincial state-owned enterprises should take up no less than 50% of the new posts to recruit fresh college graduates.
Scientific research assistant	2	2	National scientific research projects of colleges and universities and scientific research institutes give priority to employing college graduates as scientific research assistants or auxiliary personnel.

**Table 4 healthcare-10-00955-t004:** Optimizing public services.

Optimize Service	Node	Reference Point	Typical Description
Simplify the employment process	9	9	Continue to promote the decentralization of government, the combination of decentralization and management, simplify the employment process, and support more graduates to start a business and innovate.
Retain the status of a fresh student	2	2	Graduates who leave school may, in accordance with their wishes, keep their hukou and files in the school for 2 years, and take part in the examination and employment of employers as fresh graduates.
Internet + services	11	12	Optimize the public service of employment, promote the realization of online recruitment, online interview, online signing, to facilitate the docking of enterprises and students.

**Table 5 healthcare-10-00955-t005:** Regional differences.

Policy Tools	Eastern Part	Northeast China	Middle Part	West
Innovation and entrepreneurship to promote employment	5.74%	10.87%	3.12%	12.51%
Public institutions absorption	22.32%	8%	29.65%	19.19%
Improve the ability of employment	7.58%	11.09%	8.86%	0.82%
Optimizing employment services	9.83%	11.9%	7.12%	13.27%
Phased measures	1.52%	0%	0%	3.34%
Accurate help	17.77%	26.01%	27.28%	13.26%
Further study	5.41%	0.87%	8.74%	3.7%
Subsidies	29.84%	31.26%	15.23%	33.8%

## Data Availability

The data used to support the findings of this study are available from the corresponding author upon request.

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
