# Peer review of "Employment Management Policies for College Graduates under COVID-19 in China: Diffusion Characteristics and Core Issues"

_healthcare, 2022, doi:10.3390/healthcare10050955_

Round 1
Reviewer 1 Report
This article provides a descriptive evidence of employment policies implemented in China since the COVID-19 outbreak. As such, I do not believe this analysis is relevant for a journal like Healthcare.
Author Response
To the reviewer:
We were pleased to receive constructive comments from the reviewer. We appreciate a lot for all your efforts in reviewing this paper. We thank the reviewer for this suggestion and we will consider this suggestion carefully.
Reviewer 2 Report
The research problem presented in the reviewed article is interesting. The topic covers the employment management policies in the Chinese government. Firstly I suggest supplement in the title of the article, for example: "Employment Management Policies for College Graduates under COVID-19 in China. Diffusion Characteristics and Core Issues". The choice of topic is appropriate because the pandemic is officially over in Europe, but the lockdown is continued in China. Besides, 8.74 million - largest number of graduates in history, is the enormous scale of the problem. This is an unprecedented challenge in the world. Using of the diffusion theory is an interesting research procedure. I have no comments for literature review.
The Authors used the sources from different levels of administration. They showed the increase in government activity after January 2020. The conclusion "intrinsic driving force of policy formulation in the state of emergency is at the local level" - is very important. Their research shows another things too: agglomeration effect, cluster effect and proximity effect. They found the relationship between subsidy policy and promotion of employment, differences among provinces, and outlined research perspectives.
Author Response
To the reviewer:
We are very grateful to the reviewer’s comments on the manuscript, which were very encouraging. We appreciate a lot for all your efforts in reviewing this paper. These comments help us improve this paper and encourage us to plan for future studies. We thank the reviewer for the suggestion on the title, and we have revised the manuscript title based on this suggestion to make it clear.

Reviewer 3 Report
Thanks for your submission.
I felt that this was an interesting and thoughtful piece which could be enhanced through:
- the inclusion of additional literature from which a more critical review could develop;
- some further justification around the research design, the validity and sampling;
- enhanced discussion of the themes to emerge from the analysis;
- a more robust conclusion with discussion of limitations, mitigation of identified limitations and further recommendations. Good luck with your amendments.
Author Response
To the reviewer:
We were pleased to receive constructive comments from the reviewer. We appreciate a lot for all your efforts in reviewing this paper. These comments help us improve this paper and encourage us to plan for future studies. We have carefully considered the suggestions from the reviewer and revised the manuscript per the reviewers' suggestions.
Please see the attachment, we respond to the comments of the reviewer point by point.
